# The Matter of Time

**DOI:** 10.3390/e23080943

**Published:** 2021-07-23

**Authors:** Arto Annila

**Affiliations:** Department of Physics, University of Helsinki, 00014 Helsinki, Finland; arto.annila@helsinki.fi; Tel.: +358-44-204-7324

**Keywords:** the arrow of time, causality, force, free energy, natural selection, non-determinism, non-equilibrium thermodynamics, quantum, period, photon

## Abstract

About a century ago, in the spirit of ancient atomism, the quantum of light was renamed the photon to suggest that it is the fundamental element of everything. Since the photon carries energy in its period of time, a flux of photons inexorably embodies a flow of time. Thus, time comprises periods as a trek comprises legs. The flows of quanta naturally select optimal paths (i.e., geodesics) to level out energy differences in the least amount of time. The corresponding flow equations can be written, but they cannot be solved. Since the flows affect their driving forces, affecting the flows, and so on, the forces (i.e., causes) and changes in motions (i.e., consequences) are inseparable. Thus, the future remains unpredictable. However, it is not all arbitrary but rather bounded by free energy. Eventually, when the system has attained a stationary state where forces tally, there are no causes and no consequences. Since there are no energy differences between the system and its surroundings, the quanta only orbit on and on. Thus, time does not move forward either but circulates.

## 1. Introduction

Time is a big problem in physics [1,2,3]. On the one hand, we experience time passing, but the experience itself lacks a theoretical formulation such as an equation of motion. On the other hand, the laws of physics for particles, as we know them today, do not make a difference whether time flows from the past to the future or from the future to the past. However, is it not a thin line between the microscopic world of particles and the macroscopic world of our experiences? And, if so, where does the arrow of time [4] come from?

In modern physics, there is no point in even asking why things happen. In general relativity, the flow of time is without cause, so there are no consequences either. Bodies move along their optimal paths; the planets orbit the sun one cycle after the other; comets come and go. In turn, quantum mechanics does not outline alternative events but rather all possible events superposed [5]. Logically, there are parallel cosmoses since the superposition principle does not limit to the microcosm of particles [6]. 

Be that as it may, we have a hard time comprehending these theories that match but do not explain data, for an explanation usually calls for causation [7]. Einstein’s famous criticism of quantum theory, “God does not play dice”, is today deemed unwarranted. Nevertheless, the exclamation captures the crux of causality. Namely, no consequence emerges from mere chance without any proximate cause [8]. A phenomenon may appear random, but there is no guarantee that this is truly the case. Science does not have criteria for proving a phenomenon to be arbitrary. Instead, every single phenomenon in the universe should have a natural cause [9]. 

So, could it be that time does not point anywhere so long as nothing is happening? Have we simply defined the laws of physics to be independent of time? Do they so apply only to stationary-state systems? When quantities stay put, the measurement is precise as required. Conserved quantities relate to symmetries, which in turn have provided insight into the laws of physics. However, the world is unmistakably in flux.

Since the flow of time is a natural phenomenon, it seems reasonable that it, too, should be shown to have a natural cause (i.e., to be driven by forces). Thus, the flow of time should be written as an equation of motion. Such an equation of time would allow us to understand where the arrow of time comes from, why the future is unpredictable, and what gives rise to history. 

## 2. Materials and Methods

Let us employ the old empirical method to derive the equation of time. Starting from his own experience, Galileo structured observations as mathematical laws [10,11,12]. The first physics may well be a profitable approach, now that we do not even know how to tackle the problem of time. So, let us first express our own experience of time and then translate the expression into the language of physics.

A clear, frosty night under a starry sky is a great experience—except that with time it feels cold. Heat does not escape by itself and instantaneously but together with time. The observation is obvious, but that is precisely why it is precious. Can we thus infer that the passing of time always associates with a flow of energy? What is it in substance that moves when energy and time flow?

Under the starry sky, one feels cold since heat escapes from the warm skin to cold space. The experience exhibits causality. The temperature difference is the cause (i.e., force) and the loss of heat is the effect (i.e., a change in motion). The photon carries energy. But does the photon carry time too? 

In the history of science, the right question has often pointed to the answer. As Max Planck exposed in 1900, energy and period are inseparable, complementary properties of the photon [13]. However, instead of only yielding the photon energy, *E* = *hf*, from the frequency of oscillation, *f*, Planck’s constant, *h*, is the photon’s measure [14]. In the mathematically equivalent but rearranged form, *h* = *Et*, time is on the same footing as energy. As the photon wavelet propagates, time and energy move at the speed of light, *c* = λ/*t*, for all wavelengths, λ, and periods, *t* [15]. 

By this logic, time comprises periods as a trek comprises legs. This is a new viewpoint, not a new finding. In fact, the second is defined as 9,192,631,770 multiples of the period of a photon, whose energy makes the cesium-133 atom oscillate. Time, comprehended through the experience, is a tangible property of light, and is even visible; a red photon period is longer than a blue one. 

From the adopted empirical perspective, Planck’s constant is not a constant of proportionality. Instead, it is an invariant measure of the fundamental element, the quantum of action [16]. This axiomatic stance [17] would be proven false if, for example, the massless photon were to decay. The tenet would also turn out false if a photon were to split up or if energy were to stay constant in an event.

Galileo founded physics as a method for mathematizing first-hand knowledge into a universal law [10]. This instruction is what we just followed. The experience of heat escaping from the warm skin to cold space with time identifies the elemental constituent of time to the photon period. Rather than through such an experience, Planck found the constant by interlacing two equations together. While covering the whole spectrum of light, Planck’s law of radiation does not explain light. Planck was, therefore, blind to the essence of light: the photon is the carrier of time and energy. 

The proposed identification of time with the period of a quantum of action contrasts with views that regard time as an abstract, insubstantial dynamic quantity. Most notably, spacetime, amalgamating time and space into a four-dimensional manifold of general relativity, is a mathematical model. However, the spacetime abstraction does not exclude the possibility that time and energy are properties of a substance that embodies gravity [18]. For instance, a physical process, such as a running clock, takes place faster in the attic than in the basement because the two conditions differ in substance. Likewise, the rate of a chemical reaction depends on the conditions. Thus, the proposed concrete comprehension of time is not blatantly at odds with mathematical models of modern physics. But, of course, in the end, all that matters is whether the notion of time comprising the photon periods agrees with empirical evidence or not.

## 3. Results

The proposal about time comprising the periods of quanta is perhaps surprising in its simplicity. But the thought is logical. It would be confusing to consider this period and time as different concepts. They have the same unit of measure as well. Paraphrasing Leibniz, if we do not have the means to distinguish between two things, we must regard them as identical [19]. There is thus no more of a mystery hidden in time than in energy—time and energy change hand in hand. 

Now that the flow of time is associated with the flow of quanta, the equation of time can be derived from statistical mechanics of open systems. The many-body theory is posited on the axiom that everything comprises the same basic building blocks. The atomistic underpinnings date back to Ludwig Boltzmann. He understood that, not only in the case of a gas through collisions, but everything evolves through interactions toward thermodynamic balance. Similarly, Willard Gibbs theorized that compounds reach chemical equilibrium through reactions [20]. Additionally, a photon gas through interactions attains thermal equilibrium with matter, as corroborated by the black body spectrum. Accordingly, the evolution of any substance can be understood so that the quanta, the fundamental elements of everything, redistribute through all kinds of events ever more favorably in energy until the most likely state, the balance, has been attained.

### 3.1. The State Equation

Assuming that the quanta embody everything, any system can be formalized in the same way. This scale-free account in a mathematical form can be inferred from the energy level diagram (Figure 1).

Let us examine an entity indexed with *j*. Its existence can be quantified in terms of probability, _1_*P_j_* = *ϕ*_1_*ϕ*_2_*ϕ*_3_… = Π*_k_ϕ_k_*, in the form of product, Π*_k_*, over ingredients, indexed with *k*. The notation ensures that if any one component *k* is missing altogether, *ϕ_k_* = 0, then also _1_*P_j_* = 0. For example, an enzyme in a cell could not possibly exist if any one of its ingredients, say, a metal ion in the active site, were missing altogether. The power of statical mechanics stems from the fact that we can express the probability, _1_*P_j_*, even if we do not know what the components, *k*, are in the product, Π*_k_*, provided that all entities are basically made of quanta.

When the system houses several indistinguishable entities, for example, a cell houses multiple copies of an enzyme, the probability of that population, *P_j_* = [_1_*P_j_*][_1_*P_j_*][_1_*P_j_*]…/*N_j_*! = [_1_*P_j_*]*^N^**^j^*/*N_j_*! is a product of _1_*P_j_* over the size of the population, *N_j_*. Again, the product form ensures that if any one entity is missing altogether, _1_*P_j_* = 0, then also *P_j_* = 0. When the entities are identical, their mutual order makes no difference. Hence, the expression is divided by the number of ways, *N_j_*!, the entities can be arranged into a sequence. 

Finally, the total probability, *P*, of the system is the product, Π*_j_*, over *P_j_*
(1)$$P=\underset{j}{\prod }P_{j}=\underset{j}{\prod }{\left[\underset{k}{\prod }\phi_{k}\right]}^{N_{j}}/N_{j}!\;,$$
where each factor, *ϕ_k_* = *N_k_*exp[(−Δ*G_jk_* + *i*Δ*Q_jk_*)/*k_B_T*], denotes the population of starting materials, *N_k_*, and the energy differences, i.e., free energy, −Δ*G_jk_* + *i*Δ*Q_jk_*, relative to the average energy of the system, *k_B_T*. 

Since temperature, a meaningful notion for a statistical system, was taken into use long before the concept of energy, *T* is multiplied by Boltzmann’s constant, *k_B_*, to make it commensurate with the other terms of energy. When any one event, either due to absorption or emission of quanta, shifts *k_B_T* only slightly, the system evolves smoothly, as if continuously. In such a statistical system, an energy difference can be approximated using an exponential function (exp) [20,21]. The base of the natural logarithm, the limit of continuous compounding, is a natural of choice, as the function *f*(*x*) = *e^x^* is self-similar under a change, *de^x^*/*dt* = *e^x^*. 

The gap in energy, Δ*G_jk_*, between the starting material, indexed with *k*, and the product, indexed with *j*, can be bridged with the flux of energy between the system and its surroundings, Δ*Q_jk_* = *nhf_jk_*, carried by quanta with a characteristic frequency, *f_jk_*, that couple to a *jk*-transformation from the starting material into the product. The label, *i*, in front of the energy term, means that the system is open to the surroundings for the flows of quanta. For example, the influx of photons from the sun makes photosynthesis happen, and the efflux of photons from a body makes metabolism happen. The free energy expression, −Δ*G_jk_* + *i*Δ*Q_jk_*, denotes the force that an open system consumes as it evolves approximately along a logarithmic spiral, eventually settling at a closed stationary orbit, where dissipation vanishes, Δ*Q_jk_* = 0 [22].

The state equation (Equation (1)) is the main result of the non-equilibrium thermodynamic theory for open quantized systems. As shown below, a straightforward mathematical derivation starting from Equation (1) yields the equation of motion in its various forms.

The state of a system is customarily given by an additive, Σ, measure. It is obtained by taking the logarithm (ln) of the product form (Equation (1)). For historical reasons, the logarithm of probability, when multiplied by *k_B_*, is known as entropy
(2)$${S=k}_{B}{\ln P=k}_{B}\underset{j}{\sum }{\ln P}_{j}\approx \frac{1}{T}\underset{jk}{\sum }N_{j}\left({-\Delta \mu }_{jk}{+i\Delta Q}_{jk}{+k}_{B}T\right),$$
where Δ*μ_jk_* = *μ_j_* − *μ_k_* means the potential energy difference between the populations *N_k_* and *N_j_*. The population of *k*-entities embodies the potential, *μ_k_* = *k_B_T*ln*ϕ_k_*, and that of *j*-entities the potential, *μ_j_*. While the functional form of *μ_k_* is the familiar chemical potential, it is valid for any potential, assuming that everything comprises quanta. For example, an electric field potential comprises photons. The entry ≈ in Equation (2) stands for the statistical approximation, ln*N_j_*! ≈ *N_j_*ln*N_j_* − *N_j_*, which is excellent for *N_j_* > 10.

It is worth emphasizing that entropy (Equation (2)), as the logarithm of probability (Equation (1)), adds nothing to the description beyond the concept of energy. In particular, entropy is not a measure of disorder. The total energy of the system, *TS*, temperature, *T*, times entropy, *S*, comprises the system-bound energy, Σ*N_j_k_B_T*, and free energy, Σ*N_j_*(−Δ*μ_jk_* + *i*Δ*Q_jk_*) [23]. Thus, the system is subject to evolution so long as there is free energy. Conversely, at balance, where the familiar form of entropy, *S* = Σ*N_j_k_B_*, applies, all energy is bound.

### 3.2. The Equation of Motion

In a statistical system comprising numerous quanta, the quantum-by-quantum changes in populations, *N_j_*, can be conveniently denoted by differentials, *dN_j_*. Then it is easy to see that free energy terms, −Δ*μ_jk_* + *i*Δ*Q_jk_*, drive forward transformations, where *N_j_* increases. Conversely, opposing forces drive the reverse reaction, where *N_j_* decreases. As a result of *jk*-transformations, the total energy of the system, *TS*, comprising all quanta, changes with time, *t*,
(3)$$\;T\frac{dS}{dt}=T\underset{j\;}{\sum }\frac{dS}{{dN}_{j}}\frac{{dN}_{j}}{dt}=\underset{jk}{\sum }\frac{{dN}_{j}}{dt}\left({-\Delta \mu }_{jk}{+i\Delta Q}_{jk}\right).$$
As the quanta redistribute due to the gradients in energy, temperature, *T*, changes as well. However, *T* is not explicitly differentiated with respect to time because variation in the average energy follows from variation in *S*.

It is of interest that the equation of motion (Equation (3)) cannot be solved. Since Δ*μ_jk_* is a function of *N_j_*, the changes in each population, *N_j_*,
(4)$$\frac{{dN}_{j}}{dt}=\frac{1}{k_{B}T}\sum_{k}^{}\sigma_{jk}\left({-\Delta \mu }_{jk}{+i\Delta Q}_{jk}\right)\;,$$
proportional to the free energy terms by mechanism-dependent factors, *σ_jk_* > 0, cannot be separated from their driving force. In other words, the course of events is not deterministic. However, it is not random, i.e., indeterministic, either, but limited by free energy.

In the scale-free description, a mechanism, *σ_jk_*, such as an enzyme, is a system of its own. It facilitates free energy consumption by speeding up the *jk*-conversion of *N_k_* into *N_j_* or vice versa. It follows from the imperative to attain thermodynamic balance in the least time that the flows of quanta *naturally select* the most efficient mechanisms [24]. In other words, suboptimal paths dry up. It is thus the forces, i.e., free energy, at present that point to the future and transform the present (state of the system) into the past through various mechanisms.

When influxes of free energy fuel the growth, the population increases, *dN_j_*/*dt* > 0. Conversely when effluxes consume *N_j_*, the population decreases, *dN_j_*/*dt* < 0. Thus, it can be seen that entropy cannot decrease, *dS* ≥ 0, by squaring the free energy terms (Equations (3) and (4)). Note, when squaring, that these terms are orthogonal in the *jk*-basis because every motion follows its line of force, not others.

There is no exception to the second law of thermodynamics. The entropy of a system cannot decrease, not even at the expense of an increase somewhere else. The conclusion contradicts the common, yet unwarranted, understanding that an increase in entropy entails an increase in disorder. However, as apparent from the above derivation, neither the quest for order nor disorder drives the system forward but free energy consumption.

The inference about never decreasing entropy, *dS* ≥ 0, is based on the axiom that the total number of quanta is conserved. As no quanta can come out of nothingness or vanish into nothingness, the system and its surroundings coevolve toward balance so that a quantum leaving the system will end up in the environment or vice versa [22]. The conclusion is also backed up by empirical evidence. For example, both animate and inanimate systems display the same ubiquitous patterns [25,26].

When free energy may only decrease and entropy may only increase, it is the whole energy landscape, including all entities, that is in motion rather than any one entity moving on a stationary landscape. Thus, there are no energy barriers to be crossed; thermodynamics and kinetics are consistent with each other. For example, water starts to flow when the water level rises over the spillway crest. Likewise, a chemical reaction proceeds from starting materials to products when the energy of the starting materials, including chemical and kinetic energy, as well as absorbed photons, exceeds the energy of the products. Accordingly, a catalyst does not change the energy level diagram or landscape, it only speeds up the conversion of starting materials into products or vice versa. Likewise, water levels even out the faster, the larger channel. Since energy differences diminish in the least time, entropy does not just increase; it does so in the least time.

The course of events, i.e., evolution, growth, or any other change, cannot be predicted because everything depends on everything else. Nevertheless, the process can still be simulated a step at a time, according to Equation (4). Such exercises demonstrate that standards, skewed divisions, growth curves, oscillations, and chaotic courses emerge from the least-time processes [27]. In practice, the time step, *dt*, ought to be short enough not to violate the statistical approximation. It means that during *dt*, the change in free energy should not rival the bound energy, i.e., Σ*N_j_*(−Δ*μ_jk_* + *i*Δ*Q_jk_*)/*N_j_k_B_T* << 1.

### 3.3. The Continuous Equation of Motion

Although every system evolves from one state to another quantum-by-quantum, many phenomena, such as the flow of water, appear as if they were continuous motions. We obtain the continuous equation of motion from Equation (3), in terms of continuous potentials *U* and *Q* using the definitions *μ_j_* = (∂*U*/∂*N_j_*) and *Q_j_* = (∂*Q*/∂*N_j_*)
(5)$$T\frac{dS}{dt}=\sum_{jk}\frac{{dN}_{j}}{dt}\left(-\frac{\partial U}{\partial N_{j}}+\frac{\partial U}{\partial N_{k}}+i\frac{\partial Q}{\partial N_{j}}-i\frac{\partial Q}{\partial N_{k}}\right)=-\frac{\partial U}{\partial t}+i\frac{\partial Q}{\partial t}=\frac{d}{dt}2K,$$
because in the orthogonal *jk*-basis, the change, *dN_j_*, does not affect the gradient, ∂/∂*N_k_*. The change in entropy, *TdS* = *d*2*K*, translates to the change in kinetic energy because the absorption or emission of photons, carrying *Q*, causes concomitant changes in *U* and *K*. As a result, the system’s quanta assume new paths that differ from the old ones by energy and period, equivalently by momentum and wavelength. The potential energy changes per time, ∂/∂*t*, can also be written per position, ∂/∂*x_j_*, multiplied by velocity, *v_j_*, i.e., ∂*U*/∂*t* = ∑*_j_*_=*x,y,z*_ *v_j_*∂*U*/∂*x_j_* and ∂*Q*/∂*t* = ∑*_j_*_=*x,y,z*_ *v_j_*∂*Q*/∂*x_j_*.

The differential form of Equation (5) corresponds to the integral form, ∫**p** ∙ *d***x** = ∫*2Kdt*, that sums up momenta, **p**, of the quanta on their paths, **x**, or equivalently kinetic energy, 2*K*, on their periods, *t* [28]. The variation of the integral equaling zero is known as Maupertuis’ principle of least action.

Unlike Lagrangian, the original Maupertuis’ form is open for evolution. The dissipation means that the limit of integration moves during the integration because the driving forces affect the motion, affecting the forces, and so on. For this reason, the future cannot be known beforehand.

We can also obtain Equation (5) by multiplying with **v** the original form of Newton’s second law of motion and writing the change in kinetic energy, 2*K* = **v ∙ p** = ∑*v_j_mv_k_*, in the Cartesian base where the inner product vanishes for *j* ≠ *k* and *d***v**/*dt*
**∙ p** = 0 as *d***v**/*dt* ⊥ **v**, i.e.,
(6)$$F=\frac{d}{dt}p=ma+v\frac{dm}{dt}|\;v\;v\cdot F=v\cdot \frac{d}{dt}p=\frac{dx}{dt}\cdot ma+v\cdot v\frac{dm}{dt}=-\frac{dU}{dt}+i\frac{v^{2}}{c^{2}}\frac{dE}{dt}=-\frac{dU}{dt}+i\frac{dQ}{dt}\;.$$
The change in mass, *dm*/*dt* = *dE*/*c*^2^*dt* = *dQ*/*v*^2^*dt*, means, geometrically speaking, changes in curvatures of the quantized trajectories that open up and dissipate quanta into the surroundings. In this context, mass-energy equivalence, *E* = *mc*^2^, customarily understood as a relativistic formula, is motivated by extending it to the action, *Et* = *mc*^2^*t* = *px*.

Thus, the second law of thermodynamics, Maupertuis’ principle of least action, and Newton’s second law of motion are found to be one and the same law. Here the equation of non-equilibrium thermodynamics is called the equation of time [27].

It is noteworthy that the force, **F**, also contains absorbed or emitted energy, *idQ*, at the event where the system is displaced by *d***x**. The concomitant change in mass, *dm*, is big in nuclear reactions, small in chemical reactions, and always finite. In other words, masses, i.e., the geodesic curvatures of quanta, change until the system becomes stationary [29]. Poynting’s theorem is also the same law given in electromagnetic terms [22]; the work exerted by the electromagnetic forces on charges equals the change in the density of electromagnetic energy.

The quanta flow along the least-time paths, the lines of force in the words of Faraday [12]. Once the net flow of energy between the system and the surroundings has vanished, the system has attained balance in its surroundings. Then variables can be separated and trajectories computed [30]. When dissipation ceases, *dQ* = 0, the equation of motion (Equation (6)) reduces to 2*K* + *U* = 0, known as the virial theorem.

According to Noether’s theorem, 2*Kt* = *nh*, the steady-state system totals *n* quanta with kinetic energy, 2*K*. In any given stationary system, the quanta complete their full orbits within their characteristic periods, *t*; may that system be an electron torus [31,32,33] or a planet orbiting the sun. As Noether’s first theorem states, every continuous, i.e., differentiable symmetry of the action, corresponds to a conservation law. Time invariance corresponds to constant energy, translational invariance to fixed momentum, rotational invariance to fixed angular momentum. Accordingly, invariant charge, magnetic moment, and mass relate to stationary paths.

### 3.4. The Equation of Time

Since the quantum of action carries both energy and time, a change in energy relates to the progression of time, and so, the flow of time cannot but be the flow of quanta. Conversely, it would be inconsistent and confusing to think that time is something else besides the complementary property of energy in a quantum.

As the quanta flow, energy differences decrease with time. Thus, the leveling out sets the arrow of time. This irreversible motion is the essence of the second law of thermodynamics (Equations (3)–(6)). The same can be inferred from Planck’s constant, *h* = *Et*, by differentiation, *dh* = 0 ⇒ *dE*/*dt* = −*E*/*t*. Specifically, as the photon period lengthens, its energy lowers.

Since time and energy, as well as momentum and wavelength, are properties of the quantum, events in a sequence are not interchangeable. Mathematically speaking, they are noncommutative. The outcome depends on the order in which the quanta move. For example, momentum changes when measuring position. Conversely, position changes when measuring momentum. Thus, in agreement with quantum mechanics, the order of time [34] is the order in which quanta move.

The passage of time renders the universe asymmetrical in its details [2,3]. As a result, the overall distribution of matter is isotropic but not symmetrical. Genuinely symmetrical distributions emerging from random processes are found nowhere in Nature [25,26]. Even so, such distributions approximate quite well the steady-state dynamics.

Consequently, stationarity is known in precisely defined terms, such as equilibrium, conserved, commutative, computable, linear, Euclidean, and deterministic. By contrast, the full range of processes is referred to by vaguely understood antonyms, such as non-equilibrium, nonconserved, noncommutative, noncomputable, nonlinear, non-Euclidean, and non-deterministic. Thus, there is a need for a general equation of motion for nonstationary systems (i.e., an equation for non-equilibrium thermodynamics).

From the thermodynamics perspective, the rate of a process depends on the surroundings (for example, gravitational potential). In agreement with general relativity, the clock runs faster in the attic than in the basement [18,35]. Moreover, in agreement with special relativity, the speed at which the clock moves affects its rate. Dissipation decreases with speed approaching the speed of light. For example, when a spontaneously decaying particle moves very fast, almost at the speed of light, its lifespan increases greatly [36]. As dissipation is hindered, the particle cannot disintegrate that easily. Ultimately, dissipation vanishes when the energy difference to the surrounding vacuum, a light-like substance [37], narrows down to nothing. However, no particle can attain the speed of light. If it could, it would have to be like a photon, an uncuttable fundamental element, *atomos* [17].

Moreover, the optimum expressed in terms of time and energy is the same because time and energy are inseparable properties of the quantum. For example, the rotating earth’s slightly flattened form is energetically optimal, having the least-time shape. Therefore a clock runs as fast at the North Pole as at the Equator. On the one hand, the clock would run faster at the Equator than at the pole since the distance to the center of the earth is longer and, hence, gravity is weaker. On the other hand, the clock would be running slower at the Equator due to the earth’s rotation. These two opposing effects precisely cancel each other [38].

While the calculation of a stationary system, such as a closed orbit, can be precise, it is not a prediction about the future. Instead, it is a disclosure of the unknown trajectory. In such a non-dissipative system, quanta orbit closed trajectories. Since energy is conserved, also time does not advance but circulates on and on. The outcome is a paradox: the steady-state equation of motion has the elements of the explanation, but at the point of balance, where nothing happens, there are no causes or consequences to be explained [39]. The inevitable conclusion is that the future is genuinely unpredictable yet bounded by free energy. Not just anything can happen, only something for which there are forces, say, resources.

The future is not all arbitrary, even when events become chaotic (i.e., when free energy becomes comparable to the bound energy) [27]. Chaos and dramatic effects do not follow deterministically from subtle differences at the onset but non-deterministically from the tremendous forces engaged along the way, i.e., history. For example, the flap of a butterfly’s wings in Brazil does not cause a tornado in Texas [40], the temperature difference between the warm ocean and the cold upper atmosphere does. From this perspective, the tornado is a mechanism to dissipate the energy difference, not a consequence of an initial condition.

Despite the inference drawn from the equation of time, one might still suppose that if one only knew a system’s initial state exactly, any future state could also be worked out mathematically. Consider, for example, the traveling salesman problem. It is easy to think that the initial state, the starting point, the home town, is known precisely. However, as the salesman arrives at a city, the driving forces (say, travel costs) change. This change, in turn, will affect the future course, and so forth. Hence there is no effective algorithm for figuring out the least-expensive travel plan from the initial state. At worst, every possible path must be evaluated to the very end. Such a computational task is intractable (i.e., noncomputable) [30].

Also, the rationale behind the halting problem, or an undecidable problem in general, is that everything hinges on everything else. It is impossible to know a priori without executing (i.e., unleashing a flow of quanta) whether a process, such as a program with input, will finish up with output or get caught up in circulating forever.

Noncomputability is not about complexity since even problems involving only three bodies are unsolvable. The motion of one body, say, the earth affects the forces that act on the other two, say, the moon and the sun, and vice versa. Non-determinism follows from the interdependence between a system and its surroundings. Consider, for example, a rock rolling down from a hilltop to a valley. As the rock rolls, the hill height decreases and the valley bottom fills up. This motion of a landscape is perhaps not obvious in the case of one rock but apparent when the whole hill has eroded to plateau; rocks do not roll anymore.

Likewise, non-determinism presents itself in the case of a system and its observer. Even the mere act of knowing entails a flow of at least one quantum, as stated by Heisenberg’s uncertainty relation. Thus, a course of events is driven by forces (i.e., causes) rather than being random (i.e., indeterministic) without involving any forces, or being deterministic without alternatives or being deterministic in probabilistic terms among alternatives.

## 4. Discussion

For ages, the vexed question of time has preoccupied scientists and philosophers. Thus, the idea that time is the property of a quantum, such as energy, might be surprising in its simplicity and concreteness. However, we would not talk about time if it had no substance at all. And we would not talk about the arrow of time if the substance had no sense of direction as the photon has. From this perspective, a theory lacking the notion of time in substance is empirically untenable. Also, an effective theory of a sequence of events is unfalsifiable as much as it is open for amendments without substance [41]. Even so, such a mathematical model can be a good model of many processes [42].

It is of interest to contrast the proposition that consumption of free energy sets the arrow of time with the deeply rooted conviction in contemporary physics that ever-increasing disorder is what directs the flow of time [43,44]. Empirical evidence, including our own experience, is that not only disordering. Ordering also takes time. For example, we see that order increases when water freezes, and we see that disorder increases when the ice melts. Thus, both order and disorder emerge as consequences of energy differences between an environment and a system evening out [45]. It is, therefore, not an increasing disorder but an imbalance that directs the arrow of time [46].

Moreover, when a film is played backward, the course of events looks unreal. Shards of glass on the floor just cannot merge into a solid vase and rise back onto the table. For that to happen, work needs to be done, but we see no one doing it. Thus, the conclusion is that time does not step all by itself but by forces (i.e., free energy).

When time is understood like energy as a property of the quantum, the present state is the only state that exists. In other words, we can only be in the present and neither in the past nor the future [47]. This tenet, compatible with our experience, contrasts eternalism, theoretically speaking, the block universe where space and time as abstract concepts are on equal footing [48].

The realistic stance also differs from presentism since the present is understood to result from the forces present in the past. History is on display everywhere. As much as the forces (i.e., causes) are apparent, the future can be foreseen. In every case, when entropy is defined in energetic terms (Equation (3)), instead of equating it with disorder, the future will be energetically more favorable (i.e., more probable than the present) which is more probable than the past [23]. Therefore, it is only natural that the universe expands everywhere in every direction, a stone falls straight down, a plant grows toward light, and you go for the best price. In this way, free energy is consumed at the fastest rate and thermodynamic balance is attained in the shortest time. The maxim is, in a sense, a truism.

When this quest for balance in the least time is understood as *natural selection* (that is, nature selects) then evolution encompasses not just the living but everything. Temperature difference forces hot tea to cool down, just as food powers the growth of a population [49]. Be it in temperature, chemical energy, or any other difference, they all diminish by flows of quanta in the least amount of time. If orderly structures help to attain balance, they will emerge. Conversely, if disorder facilitates the process toward balance with surroundings, disorder will increase. So, increasing order or disorder is merely a consequence, not an end itself.

Long ago, the biosphere, as a mechanism in its entirety, emerged to consume the energy imbalance between matter on the globe and the hot sunlight [50]. Nowadays, solar panels gain ground for the same reason. They collect photons even more effectively than plants [51]. These transformations involve different mechanisms, *σ_jk_*, but the same underlying principle (Equations (3)–(6)). That is why the data, irrespective of scale and scope, display the same patterns [25,26], including skewed distributions, sigmoid growth curves, power laws, oscillations, and even chaos [27].

Maupertuis was taken by this holistic comprehension [52] and, apparently, also Leonhard Euler. Even though Euler had formulated the principle of least action at about the same time (1744), he defended Maupertuis against claims that Gottfried Leibniz’ formulation had preceded theirs by some 40 years [53]. Euler said about Maupertuis, “This great geometer has not only established the principle more firmly than I had done but his method, more ubiquitous and penetrating than mine, has discovered consequences that I had not obtained. After so many vested interests in the principle itself, he has shown, with the same evidence, that I was the only one to whom the discovery could be attributed” [54]. In retrospect, it might well be that Euler acknowledged Maupertuis for recognizing the principle’s non-deterministic character. In any case, Euler refuted such a principle, attributed to Leibniz, that regards both the minimum and the maximum as the optima. Despite, or more likely due to its general non-deterministic nature, the Maupertuisian action was superseded by the specific deterministic Lagrangian action.

Boltzmann sought the equation of time now derived from the statistical mechanics of open evolving systems [13]. While he was impressed by Darwin’s proposal for evolution by natural selection, he did not see the need to make a fundamental distinction between the living and the non-living and, hence, envisioned the evolution of any kind to follow the same principle. Paradoxically, Boltzmann failed to discern the dynamic as he knew the end state from deriving the expression for the balance of gas molecules. However, that stationary-state equation does not have any trace of the forces that brought about the thermodynamic balance. At the balance, nothing happens because the sum of forces is zero.

Likewise, Schrödinger’s equation is a model of a stationary system. It can be transformed into a rotating frame where time is no longer a variable. The quanta circulate on and on. Nothing happens. The model is excellent. However, it cannot deal with a change, breaking of symmetry. This variance between quantum mechanics and the 2nd law underlies the measurement problem because a measurement entails either influx or efflux of quanta.

The root of the problem with Boltzmann’s H-theorem [55] was noted by his friend Josef Loschmidt. The professor of physical chemistry wondered how an equation that is symmetric with respect to time could possibly describe the flow of time. The symmetry stems from Boltzmann modeling collisions as random processes. In the vicinity of a stationary state, it is an excellent but fundamentally flawed acausal approximation.

Furthermore, as the German mathematician Ernst Zermelo remarked, Boltzmann’s equation implies that a system that has once been in a state of imbalance would return to the same state of imbalance. Such things do not happen. The issues raised by Loschmidt and Zermelo concern likewise other equations in which energy is constant. Such equations do not explain the leveling of imbalance through flows of quanta but only model the condition of balance.

The proposal that the photon’s period is time itself is a mere trifle. Despite this evident logic, someone might still insist that time is not a physical entity but only an abstract concept, even an illusion. After all, the explanation of the arrow of time, as the flow of quanta, does not seem to invalidate the quantitative results of modern physics. However, the object here is not to contest mathematical modeling but to explain time and causality in empirical terms. Even if calculations were to remain as they largely are, the worldview does change when time is understood as concretely as energy to be the photon’s property. Similarly, the Copernican model did not immediately make it easier to calculate the orbits of planets compared with the Ptolemaic system, but the belief system was nonetheless revised.

It is difficult to break the habit of thinking that time is not a dimension. Still, there is no universal axis along which to organize all events since events occur in relation to an observer. Time is relative. The passage of time that I experience matters to me, the one you sense matters to you. Greenwich Mean Time (GMT) serves to synchronize events globally, but it is just a local convention in the universe. For example, what took place on our neighboring star, Proxima Centauri, about four years ago, is visible only here today. Time is not just what can be timed, so to speak, operational comparison. A running clock is also a system in a state of imbalance. The ticking is a series of events targeting balance; the flow of quanta embodies the flow of time.

It is pivotal that the photon is open to change since the universe could not be expanding unless the photon period was increasing and energy was decreasing. The light that departed from the blazing early universe and arrived at the cold present of our time has extended so much that our eyes cannot detect it. But as our body can still feel it, even the earliest events in the universe are not altogether beyond our range of experience. We live amidst all the history that exists. To date, the photon periods sum up to about 14 billion years from the present to the past.

Assuming that the photons are all there is, the expansion of the universe could not possibly exceed the speed of light, that is, to go beyond the unity of everything. When space stems from matter rather than nothingness, there is no fuel to power ever-faster expansion [18]. So, we may abandon the assumption that the universe could billow out ever more rapidly by dark energy. Instead, the rate of expansion, the Hubble parameter, *H* = 1/*t*, is decreasing by *d_t_H* = −1/*t*^2^, as time, *t*, is increasing [35,56]. The atomistic idea of the eternal element of everything [57] limits thus interpretations of the data on the universe’s evolution more sharply than many a contemporary model of the cosmos.

Time as the property of the photon means, for example, that light does not age in a constant vacuum, but in expanding space its period lengthens [18]. The photon experiences gravitational redshift when propagating in the expanding universe, diluting in gravity. This is apparent from the Type 1a supernovae data. While the standard cosmological model requires dark energy parameter to fit the data, the calculation by the least-time principle accounts for data without fitting [18]. The two theories are at variance since the 2nd law describes dissipative processes, whereas spacetime symmetry entails preserved properties, such as metric tensor.

Time occupied the minds of both Newton and Einstein. Today, the issue is neither absolute nor relative time but tangible time; the quantum is the matter of time. Maupertuis inferred that everything complies with his principle of least action. Could not the very least action, the quantum of action, be the ultimate basis of existence? Questions and answers intertwine. Einstein summed up the power of a worldview: it is the theory that decides what we can observe.

To see what lies in the shadows, let us illuminate reality from another angle. Let us look at the whole in terms of details and the details in terms of the whole. Let us ask what the proposed non-equilibrium thermodynamic theory of time does and does not explain. The aim is not to justify the tenet but to find out whether we understand what we see.

## Figures and Tables

**Figure 1 entropy-23-00943-f001:**
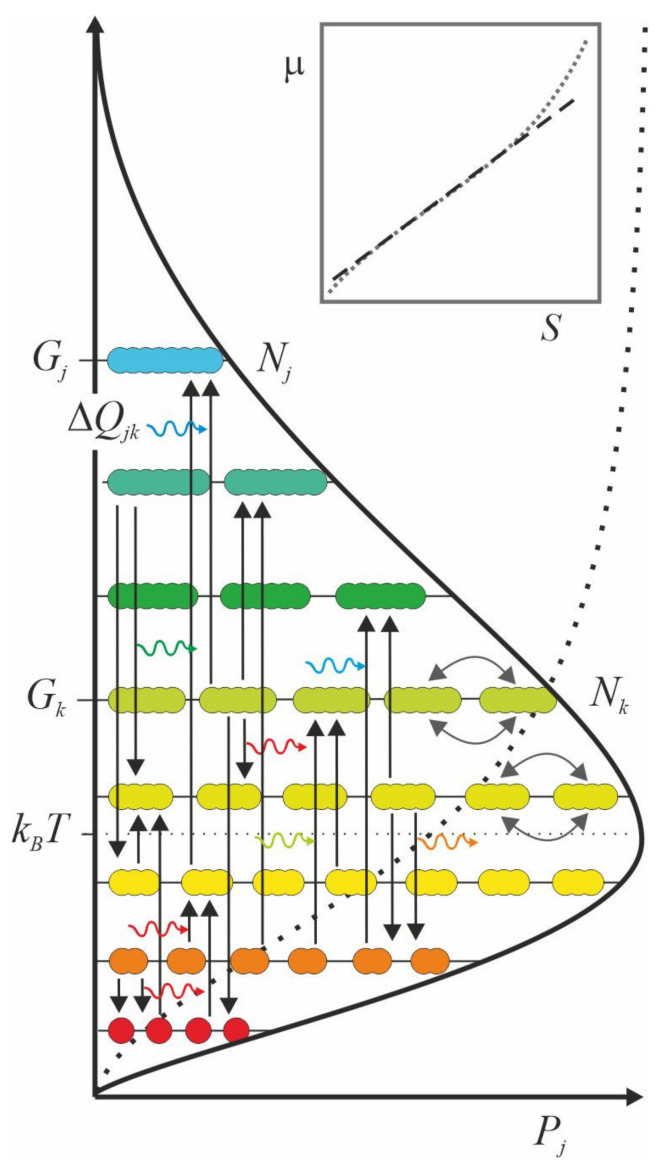
When everything comprises quanta, any system can be described by an energy level diagram. The entities of a system, in numbers *N_k_*, that have the same energy, *G_k_*, are on the same level. The bow arrows portray their mutual exchange, which changes nothing and hence causes no change in the average energy of the system, *k_B_T*, either. By contrast, the vertical arrows indicate events in which the entities move from one level to another. For example, in a chemical reaction, starting materials, *N_k_*, transform into products, *N_j_*. The horizontal wave arrows denote the quanta of light that enter the system from the environment or vice versa. Since the quanta carry energy, Δ*Q_jk_*, all events, as flows of quanta, move the system and its surroundings toward thermodynamic balance. When the energy of the surroundings is higher than that of the system, the system evolves toward higher average energy and the surrounding systems toward lower average energy, and vice versa. The cumulative probability distribution curve (dotted line) is a sigmoid. When its logarithm, entropy, *S*, is plotted as a function of (chemical) potential energy, *μ*, it mainly follows a power law, i.e., a straight line on the logarithm-logarithm scale (inset).

## Data Availability

Not applicable.

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
