# Peer review of "The Matter of Time"

_entropy, 2021, doi:10.3390/e23080943_

Round 1

Reviewer 1 Report

The paper treats a very important question and deserves to be published. Here are my general comments for its further improvement and strengthening. The ideas are very dense and the author assumes that the readers will be familiar with all of his previous ideas before they understand this paper. In the interest of the author getting more citations, becoming more popular and being understood by more people, I would suggest that he tries to give definitions and examples, so the paper is standalone and it is easier for the readers to follow, giving them more clarity. The paper may benefit from being given to more readers which can ask clarification questions, and thus it will be made more readily understood by everyone. For example the sentence “Einstein's famous criticism of quantum theory, "God does not play dice," captures the foolishness of a belief that any consequence could be the result of mere chance without any proximate cause”. This argument deserves more defense for the ideas of the author to be accepted by as many other people as possible, since it contradicts quantum mechanics and many people may be turned off by this sentence right away, as they will assume that the author uses outdated notions that have been rejected. This is like a court case which needs to be convincingly argumented and evidenced . Another example is that time stops in equilibrium systems “Eventually, when the system has attained a stationary state, where forces tally, there are no causes and no consequences. Then time does not advance as the quanta only orbit on and on.” Those are very radical statements. As we know, time rate in General Relativity depends on the stress-energy-momentum tensor and if time is to be affected by the equilibrium or disequilibrium of the system, that has to be reconciled or introduced as a correction term to prove it compatible with General Relativity. Equation 3 is an equation for non-equilibrium thermodynamics, and it will be useful for the reader to connect to that, as this may increase the readership and the citations for this article. About the “Maupertuis' principle of least action” on line 245, the variation of that integral is equal to zero is the principle, but, not the integral itself. Some sentences need to be clarified, such as “Since the quanta carry both energy and time, the flow of time cannot but be the flow of quanta.” on line 286. In general the paper will benefit from one last read, as there are a lot of punctuation errors, for example on line 311. “In every case, the future will be energetically more favorable, i.e., more probable than the present, which in turn is 382 more probable than the past [26].”,line 381, begs the explanation, how does this merge with the notion that in physical, chemical, biological and social evolution, with time, more organized disequilibrium structures appear, which, are a lot less probable than the same matter in Thermodynamic equilibrium.

Author Response

Thank you very much for investing a lot of time and energy to suggest improvements and demand corrections.

In addition to clearing the explicit concerns, in particular, backing up seemingly unwarranted statements, the manuscript is largely rewritten to make the message clear and logical.

Even after expounding, the manuscript may still fall short as a standalone. The scope is simply large. Time is involved in just about everything. In any case, all changes can be seen, as the Track Changes option has been on.

Reviewer 2 Report

I recomend paper for publication in presented form.

This is since the general problem of time which is discussed by author in reality is difficult problem for scientific discussion.It is alsp important problem in quantum mechanics where position and time are considered differently.

Author did not give final and accepted and understood to the very end answered to the problem of time.

Nevertheless for readers from time to time to read the arguments of relation of time to quanta like photons is useful to trigger the attempts of better understanding such difficult questions of nature foundations.The discussion in the paper is serious

I recommend toaccept paper as it is

Author Response

Thank you for demanding clarity and more discussion that contrasts the proposal with the contemporary stance about time. 

I have improved the manuscript accordingly. However, juxtapositioning the notion of time as the period of quantum and the concept of time in quantum mechanics and relativity theory is not exactly commensurate because the proposal is an ontological view about time whereas the view by modern physics is an instrumental one.